# The Screening Visual Complaints questionnaire (SVCq) in people with Parkinson's disease—Confirmatory factor analysis and advice for its use in clinical practice

Iris van der Lijn[1,2]*, Gera A. de Haan[1,2], Fleur E. van der Feen[1,2], Famke Huizinga[1,3], Anselm B. M. Fuermaier[1], Teus van Laar[4], Joost Heutink[1,2]

1 Department of Clinical and Developmental Neuropsychology, University of Groningen, Groningen, The Netherlands, 2 Royal Dutch Visio, Centre of Expertise for Blind and Partially Sighted People, Huizen, The Netherlands, 3 Department of General Practice and Elderly Care Medicine, University Medical Center Groningen, Groningen, The Netherlands, 4 Department of Neurology, University of Groningen, University Medical Centre Groningen, Groningen, The Netherlands

* i.van.der.lijn@rug.nl

## Abstract

### Background

The Screening Visual Complaints questionnaire (SVCq) is a short questionnaire to screen for visual complaints in people with Parkinson's disease (PD).

### Objective

The current study aims to investigate the factor structure of the SVCq to increase the usability of this measure in clinical practice and facilitate the interpretation of visual complaints in people with PD.

### Methods

We performed a confirmatory factor analysis using the 19 items of the SVCq of 581 people with PD, investigating the fit of three models previously found in a community sample: a one-factor model including all items, and models where items are distributed across either three or five factors. The clinical value of derived subscales was explored by comparing scores with age-matched controls (N = 583), and by investigating relationships to demographic and disease related characteristics.

### Results

All three models showed a good fit in people with PD, with the five-factor model outperforming the three-factor and one-factor model. Five factors were distinguished: 'Diminished visual perception–Function related' (5 items), 'Diminished visual perception–Luminance related' (3 items), 'Diminished visual perception–Task related' (3 items), 'Altered visual

**Data Availability Statement:** The Data Availability Statement is under discussion and will be provided in a forthcoming update to this article.

**Funding:** Funding for the research was received from: Visio Foundation, Amsterdam, The Netherlands (JH; https://www.stichtingnovum.org/ ), and ZonMw grant 637005001 (JH; Expertisefunctie Zintuiglijk Gehandicapten, Meerjarig deelsectorplan 2020-2022 Visueel; https://www.zonmw.nl/nl/). The funders had no role in study design, data collection and analysis, decision to publish, or preparation of the manuscript.

**Competing interests:** The authors have declared that no competing interests exist.

perception' (6 items), and 'Ocular discomfort' (2 items). On each subscale, people with PD reported more complaints than controls, even when there was no ophthalmological condition present. Furthermore, subscales were sensitive to relevant clinical characteristics, like age, disease duration, severity, and medication use.

## Conclusions

The five-factor model showed a good fit in people with PD and has clinical relevance. Each subscale provides a solid basis for individualized visual care.

## Introduction

Visual problems are highly frequent in people with Parkinson's disease (PD) [1, 2]. These problems can interfere with a wide variety of daily activities and therefore negatively affect quality of life. In addition, visual problems are predictive of poor outcomes such as anxiety, depression and dementia in people with PD [3, 4]. Unfortunately, visual problems are not always recognized in clinical practice. For example, because motor and cognitive problems are more prominent [5], or because people with neurological disorders in general may have difficulty specifying their visual problems unless structured questions are asked [6].

The recognition of visual problems in people with a neurodegenerative disorder might therefore be improved with the use of a structured self-report measure that is short (i.e. can be administered in a few minutes), suitable for use by all medical specialists, provides insight into the most common complaints, and identifies people in need of specialized eye care or rehabilitation. For this purpose, the Screening Visual Complaints questionnaire (SVCq) was developed [7].

The psychometric properties of the SVCq were evaluated by Huizinga et al. (2020) [7] in a large group of Dutch speaking participants without severe self-reported neurological, ophthalmological or psychiatric disorders (18–95 years of age). They showed that the SVCq is a psychometrically valid measure for the identification of self-reported visual complaints.

The SVCq consists of 19 items (complaint descriptions) on which participants can indicate how often they experience each complaint. Huizinga et al. (2020) [7] suggested a model of three factors, or subscales, i.e. 'Diminished visual perception' (eleven items, e.g. 'unclear vision', 'reduced contrast' or 'needing more light'), 'Altered visual perception' (six items, e.g. 'double vision', 'shaky, jerky, shifting images' or 'seeing things that others do not') and 'Ocular discomfort' (two items, i.e. 'painful eyes' and 'dry eyes'). Besides this three-factor model, a one-factor and a five-factor model showed reasonable fit in healthy individuals.

We performed a confirmatory factor analysis (CFA) on SVCq scores of a large cohort of people with PD, to examine the fit of the one-factor, three-factor and five-factor model in a clinical population. Furthermore, we compared factor scores of people with PD with scores of people without PD and related scores to several demographic and diseaserelated variables.

## Method

### Participants

A large group of people with idiopathic PD (N = 586) participated in the study. Five individuals were excluded from the analysis based on the number of missing responses (data was removed case wise if missing responses exceeded 25% of items). A control group (N = 583)

**Table 1. Demographics and disease characteristics of people with PD and age-matched controls.**

| | | People with PD | Control subjects | p-value[a] |
|---|---|---|---|---|
| N | | 581 | 583 | - |
| Sex (n, % female) | | 227, 39.1% | 214, 36.7% | .435 |
| Age (years; M ± SD) | | 69.25 ± 9.01 | 69.17 ± 8.99 | .957 |
| Education[b] (n, %) | | | | < .001 |
| | Low | 100, 17.2% | 132, 22.7% | |
| | Medium | 211, 36.3% | 146, 25.1% | |
| | High | 265, 45.6% | 303, 52.2% | |
| Disease duration (years; M ± SD) | | 7.96 ± 6.59 | - | - |
| H&Y stage (n, %) | | | | |
| | 1 | 125, 21.5% | - | - |
| | 2 | 218, 37.5% | - | - |
| | 3 | 101, 17.4% | - | - |
| | ≥4 | 49, 8.4% | - | - |
| | Missing | 88, 15.2% | - | - |
| Presence of DBS (n, %) | | 81, 13.9% | - | - |
| LEDD[c] (mg; M ± SD); missing (n, %) | | 907.75 ± 592.01; 5, 0.9% | - | - |
| Presence of severe neurological condition (n, %)[d] | | 51, 8.8%[e] | - | - |
| Presence of severe psychiatric condition (n, %)[d] | | 13, 2.2%[f] | - | - |
| Presence of any ophthalmological condition (n, %)[g] | | | | < .001 |
| | Yes | 203, 34.9% | 127, 21.8% | |
| | No | 351, 60.4% | 407, 69.8% | |
| | Unclear | 27, 4.7% | 49, 8.4% | |

*Note*: DBS = Deep Brain Stimulation; H&Y = Hoehn and Yahr staging [9]; LEDD = Levodopa equivalent daily dose; M = mean; mg = milligram; n = number;

PD = Parkinson's disease; SD = standard deviation

[a] Group differences in age were examined by a Mann-Whitney U test, and group differences in sex, educational level and presence of ophthalmological conditions by a Chi-Square test.

[b] Categorization based on the International Standard Classification of Education (ISCED) [10]

[c] LEDD calculated according to protocol of Tomlinson et al. (2010) [11]

[d] Severe conditions that were used as an exclusion criterion for control subjects and might influence vision

[e] Cerebrovascular accident (n = 16), transient ischemic attack (n = 15), epilepsy (n = 10), basilar skull fracture/traumatic injury (n = 6), thalamotomy (n = 4), encephalopathy (n = 2), brain tumor (n = 2), neuroborreliosis (n = 1), cavernoma (n = 1), and pituitary tumor resection (n = 1)

[f] Schizophrenia/psychosis (n = 13)

[g] See S1 Table

was age-matched with the remaining 581 people with PD. The matching of groups was done by splitting the PD group into age groups with a 5-year range. The number of control subjects in each age group followed the distribution of people with PD over the age groups. The largest possible number of controls in each age group was randomly selected from the total group of control subjects collected by Huizinga et al. (2020) [7]. Table 1 shows characteristics of both groups. Age and sex did not significantly differ between the groups. Level of education ($X^2$ (2, N = 1157) = 18.770, p < .001) and the presence of an ophthalmological condition ($X^2$ (1, N = 1088) = 21.280, p < .001) did, with small Cramer's *V* effect sizes, of .127 and .140, respectively [8].

## Materials

**Screening Visual Complaints questionnaire.** The SVCq [7] starts with a semi-structured inventory question asking if visual complaints are present. This question is answered on a

3-point Likert scale ('never/hardly', 'sometimes', 'often/always'). If complaints are present ('sometimes' or 'often/always'), people are asked to specify these complaints. The main body of the SVCq consists of 19 structured items, each describing a visual complaint. People rate the frequency of each complaint on the same 3-point scale. The final question of the SVCq asks about the degree of discomfort people experience in their daily lives as a result of the listed visual complaints on a scale from 0 (no discomfort) to 10 (very severe discomfort). In a Dutch community sample, the total SVCq has a good internal consistency (α = 0.85) and test-retest reliability (ICC = 0.82) [7].

## Procedure

Dutch speaking people with PD who visited a neurologist at the Parkinson Expertise Center in Groningen were asked to complete the SVCq (the Dutch version, see S1 File or https://doi.org/10.1371/journal.pone.0232232.s001; for the English version, see S2 File or https://doi.org/10.1371/journal.pone.0232232.s002), either on paper or online via Qualtrics (https://www.qualtrics.com). The questionnaires were collected between May 1, 2019 and February 3, 2021, along with demographic and disease related characteristics. All individuals were informed about the study and gave written consent for the use of their pseudo-anonymized data. According to the Medical Ethics Committee of the University Medical Center Groningen, no further ethical approval by the committee was required because all data were collected from standard care.

Data of control subjects was collected by Huizinga et al. (2020) [7] through Panel Inzicht, an online research panel in the Netherlands. A small financial reward was provided for filling out the online version of the SVCq via Qualtrics. The Ethical Committee Psychology of the University of Groningen approved this part of the data collection. All participants provided written informed consent. People in both groups could take as long as necessary to fill in the questionnaire. On average, it took about ten minutes.

## Data-analysis

LISREL 8.8 was used to perform the CFA [12]. Remaining analyzes were done in SPSS 26 (IBM Corp.) [13].

**Confirmatory factor analysis.** CFA was used as a method to determine the best fitting factor structure for the 19 SVCq items of the total sample of people with PD (N = 581). A CFA aims to test whether a relationship exists between these items and predetermined underlying factors. In this case, the factors were predetermined by an exploratory factor analysis performed by Huizinga et al. (2020) [7]. They proposed three models: a one-factor model, three-factor model, and five-factor model. Table 2 presents these models showing the items per factor.

Since a CFA is only possible with complete data, missing values in the 19 items of the SVCq (0.2% of total data) were imputed based on all available values, using the Maximum Likelihood Estimation method. Three CFAs were carried out to compare the fit of the three models for the current data. The Diagonally Weighted Least Square Method was used because of the ordinal data [14]. Scaling of latent variables was done by setting the variance of each factor to 1. The sample size exceeds the criterion of 200 participants set by Hoelter (1983) [15] for a reliable CFA.

The following goodness-of-statistics were used to assess the fit of each model. First, the Satorra Bentler Chi-square ($\chi^2$) value was determined to calculate the normed Chi-square value ($\chi^2$/df). We chose the normed Chi-square over the Chi-square value, since the Chi-square value is likely to reject models in case of large sample sizes and deviations from

**Table 2. One-factor model, three-factor model and five-factor model with belonging items as found by Huizinga et al. (2020) [7].**

| Diminished visual perception | | |
|---|---|---|
| | *Function related* | Unclear vision |
| | | Trouble focusing |
| | | Depth perception |
| | | Reduced contrast |
| | | Reading |
| | | |
| | *Luminance related* | Blinded by bright light |
| | | Needing more light |
| | | Light/dark adjustment |
| | | |
| | *Task related* | Needing more time |
| | | Looking for something |
| | | Traffic |
| | | |
| Altered visual perception | | Double vision |
| | | Shaky, jerky, shifting images |
| | | Visual field |
| | | Color vision |
| | | Seeing things that others do not |
| | | |
| | | Distorted images |
| Ocular discomfort | | Painful eyes |
| | | Dry eyes |

*Note*: The one-factor model includes all 19 items; the three-factor model consists of the factors Diminished visual perception (11 items), Altered visual perception (6 items), and Ocular discomfort (2 items); in the five-factor model, the factor Diminished visual perception is split in three factors (Function related (5 items), Luminance related (3 items), and Task related (3 items)), while the factors Altered visual perception and Ocular discomfort are the same as in the three-factor model.

normality [16], which was also demonstrated by Huizinga et al. (2020) [7]. The normed Chi-square corrects for this by taking degrees of freedom into account [17]. The normed Chi-square shows how well a model fits in comparison to no model at all. It measures the magnitude of discrepancy between the sample and fitted covariance matrices [18]. An acceptable fit is achieved when the normed Chi-square values range from 2.0 to 5.0, with lower values representing a better fit [19]. Values below 3.0 represent a good model fit. Second, the Root Mean Squared Error of Approximation (RMSEA) was determined, as well as the upper limit of the 90% confidence interval of RMSEA [16]. These measures indicate the discrepancy between the model and data covariance matrices per degree of freedom [20]. A RMSEA value less than 0.07 indicates a good model fit [21], as does a value less than 0.08 for the RMSEA confidence interval. Furthermore, the Standardized Root Mean Square Residual (SRMR) is the square root of the difference between the sample covariance residuals and the hypothesized covariance model. It ranges from 0 to 1, with values of 0.08 or lower representing good models [18]. Finally, the Comparative Fit Index (CFI) compares the sample covariance matrix with a null model, which assumes that all latent variables are uncorrelated. A CFI of 0.90 or higher is indicative of a good-fitting model [18].

To statistically compare the fit of the models, nested Chi-square tests for ordinal data were performed [22].

**Composite scale reliability.** McDonald's omega was calculated to examine the composite reliability (or internal consistency) of each factor within a model. It is an indication of the shared variance between items within a factor, which shows if items actually measure a comparable construct. The higher the shared variance, the better the reliability of the factor. The reliability is sufficient when it is greater than 0.70 [23].

*Factor scores and relationships with other variables.* Scores were calculated for each of the five factors retained from the CFA by summing the responses to the items belonging to each factor (0 = 'never/hardly', 1 = 'sometimes', 2 = 'often/always'). Since normality was violated, non-parametric tests were performed. The relationship between the subscale scores and age, disease duration, and Levodopa Equivalent Daily Dose (LEDD) was calculated by Spearman's correlations. Kruskal-Wallis and Mann-Whitney-U tests were performed to investigate differences in subscale scores between 1) people with PD and age-matched controls, 2) people with PD with and without an ophthalmological condition, 3) people with PD and controls without an ophthalmological condition, 4) male and female people with PD, and 5) people with PD in different disease severity stages (H&Y 1, H&Y 2, H&Y 3, and $\geq$ H&Y 4). An alpha smaller than .05 was considered significant. Coefficient r was calculated as an effect size (small: .1 - .3, medium: .3 - .5, large: .5–1.0) [22].

## Results

### Confirmatory factor analysis

Table 3 shows the goodness-of-fit statistics of the three models. The normed Chi-square values all fall within the range of 2.0 to 5.0, indicating a good fit. The same holds for the RMSEA (<0.07), including the upper limit of its CI (<0.08), the SRMR ($\leq$0.08), and the CFI (>0.90) of all models. Goodness-of-fit statistics show that the three-factor model had a better fit than the one-factor model, and the five-factor model had a better fit than the three-factor model and the one-factor model. Nested Chi-square tests supported this finding. Significant differences were found between the one-factor model and the three-factor model ($\chi^2$ (3, 581) = 31.30, p < .001), the one-factor model and the five-factor model ($\chi^2$ (10, 581) = 304.63, p < .001), and the three-factor model and the five-factor model ($\chi^2$ (7, 581) = 167.80, p < .001).

### Composite scale reliability

The composite reliability of each factor within the three models is presented in Table 4. The complete SVCq, or one-factor model, showed good reliability. In both the three-factor model and the five-factor model, the reliability was good for all factors except for the 'Ocular discomfort' factor in both models and the 'Luminance related' factor in the five-factor model.

**Table 3. Goodness-of-fit statistics of the one-factor model, the three-factor model and the five-factor model in the PD sample.**

| Model | $\chi^{2a}$ (df) | $\chi^{2a}$/df | RMSEA | CI-RMSEA | SRMR | CFI |
|---|---|---|---|---|---|---|
| 1 factor | 436.56 (152) | 2.87 | 0.057 | 0.063 | 0.071 | 0.99 |
| 3 factors | 345.56 (149) | 2.32 | 0.048 | 0.054 | 0.059 | 0.99 |
| 5 factors | 281.59 (142) | 1.98 | 0.041 | 0.048 | 0.054 | 0.99 |

*Note*: PD = Parkinson's disease; $\chi^2$ = Chi-square; df = degrees of freedom; $\chi^2$/df = normed Chi-square; RMSEA = Root Mean Squared Error of Approximation; CI = confidence interval (upper limit); SRMR = Standardized Root Mean Square Residual; CFI = Comparative Fit Index.

[a] Satorra-Bentler Scaled Chi-Square

**Table 4. Composite reliability of each factor within the three models.**

| Model | Factor (N items) | Composite reliability (ω) [a] |
|---|---|---|
| 1 factor | Visual complaints (19) | .90* |
| 3 factors | Diminished visual perception (11) | .89* |
| | Altered visual perception (6) | .72* |
| | Ocular discomfort (2) | .57 |
| 5 factors | Diminished visual perception–Function (5) | .83* |
| | Diminished visual perception–Luminance (3) | .69 |
| | Diminished visual perception–Task (3) | .77* |
| | Altered visual perception (6) | .72* |
| | Ocular discomfort (2) | .57 |

[a] McDonald's omega cannot be calculated for two-item scales. Therefore, the Spearman-Brown coefficient was used for the 'Ocular discomfort' subscale [24].

* good composite reliability [23].

## Factor scores and relationships with other variables

Since the five-factor model showed the best fit, all subsequent results are based on this model. People with PD reported significantly more complaints than control subjects on all subscales (see Table 5). Effect sizes were small.

Table 6 shows that people with PD with an ophthalmological condition reported significantly more complaints than those without an ophthalmological condition on all subscales. In addition, a significant difference was found between people with PD without an ophthalmological condition and control subjects without an ophthalmological condition. This was found for all subscales, except the 'Ocular discomfort' subscale (i.e. painful and dry eyes). Effect sizes for all comparisons were small.

Table 7 presents results of male and female people with PD. Females reported more complaints regarding 'Ocular discomfort' compared to males. In contrast, males experienced more complaints regarding luminance ('Diminished visual perception—Luminance'; i.e. blinded by bright light, needing more light, and light/dark adaptation). Scores on other subscales did not differ between the sexes. Effect sizes were small.

The Kruskal-Wallis test performed on scores of people with PD in different H&Y stages showed that all subscales, except the 'Ocular discomfort' subscale, differed significantly between the groups (see Table 8). The Mann-Whitney U test comparing individual groups showed multiple significant differences (see Table 9). All differences found indicate that people

**Table 5. Subscale scores of people with PD and the control group, with Mann-Whitney U test results.**

| | People with PD (n = 581) | | Control subjects (n = 583) | | | | |
|---|---|---|---|---|---|---|---|
| | M ± SD | Median | M ± SD | Median | U | p | r |
| Diminished visual perception—Function | 3.36 ± 2.85 | 3.00 | 2.35 ± 2.18 | 2.00 | 137449.5 | < .001* | 0.17 |
| Diminished visual perception—Luminance | 1.71 ± 1.69 | 1.00 | 1.26 ± 1.44 | 1.00 | 145411.5 | < .001* | 0.13 |
| Diminished visual perception—Task | 1.22 ± 1.61 | 1.00 | 0.50 ± 0.94 | 0.00 | 126344.0 | < .001* | 0.25 |
| Altered visual perception | 1.44 ± 1.99 | 1.00 | 0.58 ± 1.23 | 0.00 | 122275.0 | < .001* | 0.27 |
| Ocular discomfort | 0.70 ± 0.97 | 0.00 | 0.50 ± 0.79 | 0.00 | 154304.5 | .003* | 0.09 |

*Note*: M = mean; n = number; PD = Parkinson's disease; SD = standard deviation.

* = significant p-value (α < .05)

**Table 6. Subscale scores of people with and without an ophthalmological condition, with Mann-Whitney U test results.**

| | PD OC+ (n = 203) | | PD OC- (n = 351) | | Control OC- (n = 407) | | PD OC+ vs. PD OC- | | | PD OC- vs. Control OC- | | |
|---|---|---|---|---|---|---|---|---|---|---|---|---|
| | M ± SD | Median | M ± SD | Median | M ± SD | Median | U | p | r | U | p | r |
| Diminished visual perception—Function | 3.87 ± 3.11 | 3.00 | 3.04 ± 2.68 | 3.00 | 2.20 ± 2.05 | 2.00 | 30501.0 | .004* | 0.16 | 59803.0 | < .001* | 0.17 |
| Diminished visual perception—Luminance | 2.09 ± 1.86 | 2.00 | 1.51 ± 1.58 | 1.00 | 1.13 ± 1.32 | 1.00 | 29250.5 | < .001* | 0.16 | 62958.5 | .003* | 0.13 |
| Diminished visual perception—Task | 1.55 ± 1.80 | 1.00 | 1.06 ± 1.49 | 0.00 | 0.41 ± 0.80 | 0.00 | 30318.5 | .002* | 0.15 | 54501.0 | < .001* | 0.27 |
| Altered visual perception | 1.89 ± 2.31 | 1.00 | 1.19 ± 1.76 | 0.00 | 0.44 ± 0.89 | 0.00 | 28530.5 | < .001* | 0.17 | 53940.5 | < .001* | 0.26 |
| Ocular discomfort | 0.93 ± 1.12 | 1.00 | 0.53 ± 0.83 | 0.00 | 0.42 ± 0.72 | 0.00 | 28759.0 | < .001* | 0.20 | 67476.0 | .113 | 0.07 |

*Note*: M = mean; n = number; OC+ = people with an ophthalmological condition; OC- = people without an ophthalmological condition; PD = Parkinson's disease; SD = standard deviation.

* = significant p-value (α < .05)

with PD in higher H&Y stages experienced more complaints. There were no differences between H&Y stage 1 and 2. Comparisons between other groups (1 vs. 3, 2 vs. 3, and 3 vs. ≥ 4) all revealed some significant differences. The comparisons 1 vs. ≥ 4 and 2 vs. ≥ 4 revealed significant differences for all subscales. The 'Altered visual perception' subscale (e.g. double vision or seeing things that others do not) showed significant differences in all comparisons (except 1 vs. 2). Most effect sizes were small. Medium effect sizes were found for comparisons of the groups 1 and 2 with group ≥ 4 for the 'Altered visual perception' subscale, and for the comparison between group 1 and ≥ 4 for the 'Diminished visual perception—Function' subscale.

Age, disease duration, and LEDD showed significant positive correlations with most subscales (see Table 10). Exceptions were the relationship of age with 'Diminished visual perception—Function', and 'Diminished visual perception—Luminance', and the relationship of disease duration and LEDD with 'Ocular discomfort'. Correlations were all weak [25].

## Discussion

The SVCq was developed to screen for visual complaints in people with neurodegenerative disorders, including PD. Huizinga et al. (2020) [7] evaluated the psychometric properties and factor structure of the SVCq in a Dutch community sample. The current study aimed to confirm this structure in people with PD, in order to use the subscales of the questionnaire in clinical practice and to optimize the interpretation of the presented visual complaints in people with PD.

Our study showed that each of the models with a reasonable or good fit in a community sample also has a good fit in people with PD. This means that items in each factor within each

**Table 7. Subscale scores of male and female individuals with PD, with Mann-Whitney U test results.**

| | Males with PD (n = 354) | | Females with PD (n = 227) | | | | |
|---|---|---|---|---|---|---|---|
| | M ± SD | Median | M ± SD | Median | U | p | r |
| Diminished visual perception—Function | 3.24 ± 2.87 | 3.00 | 3.55 ± 2.82 | 3.00 | 37401.0 | 0.156 | 0.06 |
| Diminished visual perception—Luminance | 1.72 ± 1.69 | 1.00 | 1.68 ± 1.70 | 1.00 | 35978.5 | 0.025* | 0.09 |
| Diminished visual perception—Task | 1.17 ± 1.59 | 0.00 | 1.30 ± 1.63 | 1.00 | 37509.0 | 0.148 | 0.06 |
| Altered visual perception | 1.52 ± 2.06 | 1.00 | 1.32 ± 1.85 | 1.00 | 38979.0 | 0.521 | 0.03 |
| Ocular discomfort | 0.69 ± 0.98 | 0.00 | 0.71 ± 0.97 | 0.00 | 35733.0 | 0.011* | 0.11 |

*Note*: M = mean; n = number; PD = Parkinson's disease; SD = standard deviation.

* = significant p-value (α < .05)

**Table 8. Subscale scores of people with PD in different disease severity stages, with Kruskal-Wallis test results.**

| | H&Y 1 (n = 125) | | H&Y 2 (n = 218) | | H&Y 3 (n = 101) | | H&Y ≥ 4 (n = 49) | | | | |
|---|---|---|---|---|---|---|---|---|---|---|---|
| | M ± SD | Median | M ± SD | Median | M ± SD | Median | M ± SD | Median | H | df | p |
| Diminished visual perception—Function | 2.99 ± 2.92 | 2.00 | 3.17 ± 2.69 | 3.00 | 3.48 ± 2.50 | 3.00 | 5.34 ± 2.96 | 5.00 | 25.94 | 3 | < .001* |
| Diminished visual perception—Luminance | 1.70 ± 1.72 | 1.00 | 1.73 ± 1.67 | 1.00 | 1.61 ± 1.74 | 1.00 | 1.57 ± 1.43 | 1.00 | 11.91 | 3 | .008* |
| Diminished visual perception—Task | 0.95 ± 1.47 | 0.00 | 1.27 ± 1.64 | 1.00 | 1.11 ± 1.33 | 1.00 | 1.35 ± 1.79 | 1.00 | 19.51 | 3 | < .001* |
| Altered visual perception | 1.08 ± 1.87 | 0.00 | 1.19 ± 1.75 | 1.00 | 1.72 ± 1.98 | 1.00 | 2.70 ± 2.49 | 2.00 | 36.52 | 3 | < .001* |
| Ocular discomfort | 0.68 ± 0.90 | 0.00 | 0.71 ± 1.03 | 0.00 | 0.55 ± 0.86 | 0.00 | 0.57 ± 0.89 | 0.00 | 1.34 | 3 | .719 |

*Note*: H&Y = Hoehn and Yahr staging [9]; M = mean; n = number; PD = Parkinson's disease; SD = standard deviation.

* = significant p-value (α < .05)

**Table 9. Subscale scores of people with PD in different disease severity stages, with Mann-Whitney U test results.**

| | H&Y 1 vs. H&Y 2 | | | H&Y 1 vs. H&Y 3 | | | H&Y 1 vs. H&Y ≥ 4 | | | H&Y 2 vs. H&Y3 | | | H&Y 2 vs. H&Y ≥ 4 | | | H&Y 3 vs. H&Y ≥ 4 | | |
|---|---|---|---|---|---|---|---|---|---|---|---|---|---|---|---|---|---|---|
| | U | p | r | U | p | r | U | p | r | U | p | r | U | p | r | U | p | r |
| Diminished visual perception—Function | 12692.0 | .286 | 0.06 | 5348.0 | .046* | 0.13 | 1709.0 | < .001* | 0.35 | 10015.5 | .191 | 0.07 | 3150.5 | < .001* | 0.28 | 1597.5 | < .001* | 0.29 |
| Diminished visual perception—Luminance | 12848.5 | .364 | 0.05 | 5922.0 | .413 | 0.05 | 2377.5 | .019* | 0.18 | 9620.5 | .062 | 0.10 | 3793.0 | .001* | 0.20 | 2014.5 | .060 | 0.15 |
| Diminished visual perception—Task | 13402.0 | .784 | 0.02 | 5703.0 | .185 | 0.09 | 2061.5 | < .001* | 0.27 | 9680.0 | .064 | 0.10 | 3422.5 | < .001* | 0.26 | 1802.0 | .005* | 0.23 |
| Altered visual perception | 12630.0 | .222 | 0.07 | 4742.0 | .001* | 0.23 | 1593.0 | < .001* | 0.39 | 9001.0 | .006* | 0.15 | 3032.5 | < .001* | 0.30 | 1830.0 | .008* | 0.22 |

*Note*: H&Y = Hoehn and Yahr staging [9]; PD = Parkinson's disease.

* = significant p-value (α < .05)

**Table 10. Spearman's correlations between subscale scores and age or disease duration.**

| | Age | Disease duration | LEDD |
|---|---|---|---|
| Diminished visual perception—Function | r = -.061, p = .140 | r = .220, p = < .001* | r = .241, p = < .001* |
| Diminished visual perception—Luminance | r = .080, p = .054 | r = .085, p = .041* | r = .171, p = < .001* |
| Diminished visual perception—Task | r = .101, p = .015* | r = .135, p = .001* | r = .189, p = < .001* |
| Altered visual perception | r = .126, p = .002* | r = .237, p = < .001* | r = .237, p = < .001* |
| Ocular discomfort | r = .097, p = .019* | r = .066, p = .113 | r = .003, p = .942 |

*Note*: LEDD = Levodopa equivalent daily dose

* = significant p-value (α < .05)

model seem to significantly relate to the same underlying construct. Therefore, the use of either model would be justified. However, the three-factor model and the five-factor model outperformed the one-factor model. So, instead of calculating a total SVCq score with all 19 items, it is valuable to calculate either three or five subscale scores when administering the SVCq to people with PD.

Arguments for using the three-factor model would be that in general, simple models are preferable to complex models [26], and the results are consistent with those of Huizinga et al. (2020) [7]. Also, in the three-factor model, only one factor showed lower composite reliability, while in the five-factor model, two factors showed lower composite reliability.

However, goodness-of-fit statistics showed that the five-factor model provides an even better fit than the three-factor model in people with PD. This was supported by the nested Chi-

square test results. Moreover, the marginally lower composite reliability in this model can be explained by the lower number of items per factor (see Table 4), as composite reliability is likely to decrease as fewer items are included [23, 27]. The smaller number of items per factor is a result the division of the factor 'Diminished visual perception', which consists of eleven complaints in the three-factor model, and is split in three subfactors in the five-factor model: 'Function related' (5 items; e.g. unclear vision and reduced contrast), 'Luminance related' (3 items; e.g. blinded by bright light and needing more light), and 'Task related' (3 items; e.g. looking for something and traffic).

Analyses on the five-factor model showed that the division in five subscales has clinical merit. Besides the fact that items in each of the factors are clearly statistically related to the same underlying construct, the structure also made sense clinically. The concordance between items in each factor was clear, making it easy for us to name the factors (e.g. blinded by bright light, needing more light, and light/dark adjustment all clearly relate to luminance conditions). Additional clinically relevant factors of this five-factor model allow for a more detailed interpretation of a patient's complaints and a clearer focus of treatment.

While using the five-factor model in people with PD, we found that the most frequent complaints were present in the function and luminance related subscales ('Diminished visual perception—Function' and 'Diminished visual perception—Luminance'). Complaints belonging to other subscales ('Diminished visual perception—Task', 'Altered visual perception', and 'Ocular discomfort') were less prevalent (see Table 5). These findings were consistent with previous findings of Borm et al. (2020) [28]. In their study the two most common complaints in people with PD were also related to either visual functions (i.e. 'I have blurry vision when I read or work on a computer') or luminance (i.e. 'When I drive at night, the oncoming headlights cause more glare than before'), while complaints regarding other subscales were less prevalent (e.g. 'I have double vision', which relates to 'Altered visual perception' in our study or 'I have a burning sensation or gritty feeling in my eyes', which relates to 'Ocular discomfort' in our study). This pattern was not unique for people with PD, since it was also present in control subjects. But, even though the pattern of complaints was similar, we found that people with PD did report significantly more complaints on each subscale compared to controls.

Complaints on all subscales, except the 'Ocular discomfort' subscale, can be explained by both the presence of ophthalmological conditions, and other factors likely related to the pathophysiology of PD, like retinal problems or an impaired visual processing [29]. This is supported by the fact that even if there is no underlying ophthalmological condition present, people with PD still experience visual complaints.

Most subscale scores were positively related to age, disease duration, disease severity, and LEDD. However, the 'Ocular discomfort' subscale is an exceptional subscale, because it is not influenced by disease duration, severity or LEDD. In addition, the difference between people with PD and controls in the 'Ocular discomfort' scale is mainly explained by the presence of ophthalmological conditions. Some ophthalmological conditions, like blepharitis, meibomian gland disease, or decreased tear production, often co-occur with PD [30]. Especially in combination with a reduced blink rate in PD, this may cause dry or painful eyes, consistent with the items of the 'Ocular discomfort' scale [31]. Attention to these complaints is highly relevant, since these might be well treated or relieved by an ophthalmologist (e.g. by artificial tears, eyelid hygiene, or warm compresses) [32].

## Clinical implications

The results of our study suggest that it is relevant to distinguish five subscales in the SVCq for a thorough interpretation of visual complaints in people with PD. Complaints in each of the

subscales might result in different daily life problems, which lead to different targets in care or rehabilitation. Furthermore, the underlying cause of complaints in each factor may be different, and relevant to address in visual care. For example, in case of complaints related to ocular discomfort, there should be attention to a possible underlying ophthalmological condition. People might also experience complaints related to luminance, for example light sensitivity due to cataract or other ocular media opacities. These people may be helped by wearing filtered glasses [33]. Others may be advised more task lighting [34].

Not all factors exhibited optimal reliability. Therefore, one should be cautious relying solely on the subscale scores. The total score of the SVCq is a valuable measure to get a straightforward picture of the overall degree of visual complaints. The high reliability of the total questionnaire (one-factor model) supports the use of the total SVCq score. Moreover, scores on individual items might provide additional insight into the specific targets for care or rehabilitation.

Other results to be aware of in clinical practice, are that even when there is no underlying ophthalmological condition, people with PD experience more visual complaints than control subjects. Factors directly or indirectly related to the disease can lead to these visual complaints. We showed that some of these factors were age, disease duration, disease severity, and the amount of medication used (LEDD). We can therefore conclude that visual complaints seem to increase as the disease progresses. Therefore, regular screening for visual complaints in people with PD is advised, even if no known ophthalmological condition is present. Regular screening results in early detection of visual complaints, which may subsequently lead to more optimal care and rehabilitation, preventing further worsening of visual complaints and associated poor outcomes, such as anxiety, depression, and dementia.

## Strengths, limitations and recommendations for future research

The current study used a large dataset of SVCq completed by people with PD. This was an outpatient group, meaning that the patients were not bedridden and thus unlikely to be in the final disease stages. Therefore, the results of this study apply only to people attending an outpatient clinic. Nonetheless, outpatients are the original target population of the SVCq. These people, and not people in later PD disease stages, are able to rehabilitate and will benefit most from rehabilitation. The large sample size in this study contributes to the representativeness of the outpatient group and the reliability of the results.

We cannot rule out that comorbidities (e.g. ophthalmological, neurological or psychiatric conditions) explain part of the complaints experienced by people with PD. By allowing comorbidities, however, we did create a representative group of people with PD. Furthermore, we do not expect that excluding comorbidities in the PD group would have led to different results in terms of factor structure, as the model fit we found here was in fact very similar to that of controls without severe comorbidities. In the analyses on subscale scores, we chose to investigate the influence of ophthalmic disorders. Future research could focus on the influence of neurological and psychiatric disorders on each of the subscales.

Our study exclusively focused on people with idiopathic PD and did not include people with other types of parkinsonism. Since different types of parkinsonism have different visual symptom expressions [35], future research might focus on other types of parkinsonism in order to provide care guidelines for these patient groups as well.

This factor analysis performed on data from people with PD, is an important step in providing guidelines for the use of the SVCq in clinical practice. To complete the validation of the SVCq in a clinical sample, future research might focus on convergent and divergent validity, and test-retest reliability of the SVCq in people with PD. Furthermore, the English version of

the questionnaire has yet to be validated. Nevertheless, the SVCq has already proven to be a well-designed and relevant questionnaire for use in clinical practice, as the SVCq was found to be a psychometrically valid and reliable measure in a community sample [7] and initial results from clinical samples are consistent with these findings (i.e. our study and a study in people with multiple sclerosis [36]).

## Conclusion

The CFA showed that, in people with PD, the SVCq is best divided into five subscales: 'Diminished visual perception—Function', 'Diminished visual perception—Luminance', ' Diminished visual perception—Task', 'Altered visual perception', and 'Ocular discomfort'. Each of these subscales contributes to the understanding of a person's complaints. In turn, this may guide the best type of treatment, as complaints on each subscale may be best addressed by other types of visual care or rehabilitation. To prevent unnecessary poor outcomes and reduced quality of life, regular screening of visual complaints using the SVCq is recommended, as visual complaints seem to increase with disease progression and are not always determined by the presence of an underlying ophthalmological condition.

## Supporting information

**S1 Table. Ophthalmological conditions in people with PD and age-matched controls.** (PDF)

**S1 File. Screening Visual Complaints questionnaire–Dutch version.** (PDF)

**S2 File. Screening Visual Complaints questionnaire–English version.** (PDF)

## Author Contributions

**Conceptualization:** Iris van der Lijn, Gera A. de Haan, Fleur E. van der Feen, Famke Huizinga, Anselm B. M. Fuermaier, Teus van Laar, Joost Heutink.

**Data curation:** Iris van der Lijn, Gera A. de Haan, Teus van Laar.

**Formal analysis:** Iris van der Lijn.

**Investigation:** Iris van der Lijn.

**Methodology:** Iris van der Lijn, Gera A. de Haan, Fleur E. van der Feen, Famke Huizinga, Anselm B. M. Fuermaier, Teus van Laar, Joost Heutink.

**Project administration:** Iris van der Lijn.

**Supervision:** Gera A. de Haan.

**Writing – original draft:** Iris van der Lijn.

**Writing – review & editing:** Iris van der Lijn, Gera A. de Haan, Fleur E. van der Feen, Famke Huizinga, Anselm B. M. Fuermaier, Teus van Laar, Joost Heutink.

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
