## [Decision Letter · Decision Letter 0]

6 Jul 2022

PONE-D-22-10017The Screening of Visual Complaints questionnaire (SVCq) in people with Parkinson’s disease - Confirmatory factor analysis and advice for its use in clinical practicePLOS ONE

Dear Dr. van der Lijn,

Thank you for submitting your manuscript to PLOS ONE. We apologise for the delay in getting this response back to you, owing to many potential Reviewers being unavailable. After careful consideration, we feel that it has merit but does not fully meet PLOS ONE’s publication criteria as it currently stands. Therefore, we invite you to submit a revised version of the manuscript that addresses the points raised during the review process.

In the Revision I would encourage the authors to consider a more structured approach to presenting the findings. The current version made it difficult to appropriately understand the results section. This should ideally mirror more closely the Methods. However, there are substantial gaps in how the Methods are described so this section also requires further detail.

We look forward to receiving your revised manuscript.

Kind regards,

Diego Kaski, PhD MBBS

Academic Editor

PLOS ONE

Journal Requirements:

Reviewers' comments:

Reviewer's Responses to Questions

**Comments to the Author**

1. Is the manuscript technically sound, and do the data support the conclusions?

Reviewer #1: Partly

2. Has the statistical analysis been performed appropriately and rigorously? 

Reviewer #1: Yes

3. Have the authors made all data underlying the findings in their manuscript fully available?

Reviewer #1: No

4. Is the manuscript presented in an intelligible fashion and written in standard English?

Reviewer #1: Yes

5. Review Comments to the Author

Reviewer #1: This manuscript examined a previously validated Visual Complaints Questionnaire (SVCq) and investigated how this applies to people with Parkinson’s disease. They recruited a large sample of people with PD (n=581 included in the analyses) plus n= 583 controls. They examined the questionnaire as applied using 3 different models (5-factor, 3-factor, and 1-factor). They also examined the application of the questionnaire in PD. They showed that overall, people with PD had more visual complaints, as assessed using the SVCq, even when excluding people with ophthalmic disease. They also showed an effect of age and disease duration.

There is increasing interest in examining visual dysfunction in people with PD, especially as this is increasingly shown to relate to poorer outcomes. The authors are also to be commended for referring to “people with PD” rather than “patients”.

However, I felt that the way that the manuscript was organised was confusing, and that more clarification was needed in several areas to help the reader understand how the questionnaire was analysed; about the origin of the patients and controls in this study; and most importantly why this all matters and how this work helps us understand PD better.

Major comments

1. I found the Methods section confusing to follow. Could the authors please explain exactly how they performed the factor analyses? Were these simply the sum of the questions in each of the sections? The organization of the Methods section could be more logical. For example, a clearer explanation of the factor analysis early on, and handling of missing data and summary of statistical analyses towards the end.

2. Can we please have some more information about the control participants, so that we can understand whether they were reasonably matched. For example, patients attending a PD clinic and controls filling-in an online form for financial reward may have very different levels of educational and socio-economic background. It might make sense to also ask spouses of patients to complete these questionnaires to get better matching of the groups.

3. The ethical framework for the patient group states that patients gave written consent for their anonymized data to be used. There is a statement that no ethical approval is required as data were collected from standard care. That may be the case for the patients, but can the authors please clarify what was the ethical framework for the controls?

4. I note that patients with other neurological and psychiatric conditions were included here. These should be excluded from this type of analysis, as other neurological and psychiatric conditions may confound the main question of interest.

5. I also found the results section written in a confusing way. It was not made clear why it mattered that each of the models were a good fit for the data, or what good reliability means for this dataset.

6. Further to this, some of the more interesting findings were buried towards the end of the results section. For example, in the PD group, even in patients with no ophthalmic diagnoses, scores were higher on the SVCq. This could be brought out more.

7. It is important that the questionnaire scores increased with age. This should be corrected for in the main analyses.

8. I also note Plos One policy that data presented in manuscripts should be made openly available. The current statement is that data will be shared upon reasonable request. In order to comply with plos one policy, I would suggest that some form of the data presented here should be made openly available.

Minor comments

1. Figure 1 is actually a list of questions from the questionnaire and should be changed to a table.

2. The abstract is quite hard to follow. Please can the authors clarify what they mean by the different factor models and why this is important to test, in the abstract.

3. Introduction, Page 3, line 41. It is worth mentioning that visual changes are often subtle, and that patients do not always mention them unless specifically asked. And also that the reason it is worth looking for these is that visual changes are predictive of poor outcomes in PD (see e.g. Hamedani et al 2020; Anang 2014).

4. Table 1: please can the authors add comparative statistics and p values as additional columns.

5. It would be good to have the full questionnaire as a table or supplemental element so that the reader can easily find this. Also please can you add information on how long it takes to administer.

6. Page 5, line 98 – were data anonymized or pseudo-anonymised?

7. Please can acronyms e.g. RMSEA, CFA, be explained at first use?

8. Page 8, line 155 “statistics show” – please clarify which statistics.

9. Table 2. Which column is the normed chi square?

10. Tables 4,5, and 6, were these all the 5 factor model?

11. Page 15, line 247, the findings of Borm et al, were these in PD or in controls?

12. Could mention in the Discussion that these questionnaires help identify patients with PD who have poor visual function and that this could be used to stratify patients and identify those at higher risk for poor outcomes such as dementia.

6. PLOS authors have the option to publish the peer review history of their article (what does this mean?). If published, this will include your full peer review and any attached files.

Reviewer #1: No

---

## [Author Response · Author response to Decision Letter 0]

15 Jul 2022

July 15, 2022

Dear Dr. Emily Chenette and reviewer, 

Enclosed you will find our revised original research article. First of all, we would like to thank you kindly for your attentive and comprehensive feedback. Below we describe the adjustments we made. We have done our utmost to incorporate the comments as well as possible. We have paid particular attention to the method and results section and have ensured that the structure of both sections is consistent. We have also complemented the method section to ensure repeatability. In addition, we attached our data for public availability. We believe that these changes will bring much clarity. Thank you for considering our revised manuscript.

Sincerely yours,

On behalf of Prof. Teus van Laar, neurologist

Iris van der Lijn, MSc, PhD candidate

Royal Dutch Visio & University of Groningen

Department of Clinical and Developmental Neuropsychology 

Grote Kruisstraat 2/1

9712 TS Groningen

The Netherlands

+316-36319906

i.van.der.lijn@rug.nl

Comments of the reviewer

Major comments 

1. I found the Methods section confusing to follow. Could the authors please explain exactly how they performed the factor analyses? Were these simply the sum of the questions in each of the sections? The organization of the Methods section could be more logical. For example, a clearer explanation of the factor analysis early on, and handling of missing data and summary of statistical analyses towards the end. 

 Under the subheading “Data-analysis” in the Method section we both clarified what a factor analysis is, as well as what the exact aim of the currently used confirmatory factor analysis (CFA) is. In addition, we have explained more about how a CFA works: mainly that a CFA investigates relationships between items and the predetermined underlying factors found by Huizinga et al. (2020). We have also relocated the paragraph on missing data, which in our opinion makes the structure of the method section clearer.

2. Can we please have some more information about the control participants, so that we can understand whether they were reasonably matched. For example, patients attending a PD clinic and controls filling-in an online form for financial reward may have very different levels of educational and socio-economic background. It might make sense to also ask spouses of patients to complete these questionnaires to get better matching of the groups. 

 We explained the age-matching procedure in more detail under the subheading “Participants” in the method section. As now stated, data on controls was collected by Huizinga et al. (2020) and we selected controls from this group based on age. 

Asking spouses of patients as controls could be a convenient way of collecting data for the control group. As a large proportion of people with PD are men, partners will in most cases be women. It is also a fact that women are, on average, less educated than men in this age category in the Netherlands. Therefore, asking spouses would have created a larger difference between the groups than is currently the case. Furthermore, asking spouses would make the two samples less independent in the current situation.

To give further insight into the similarities and differences between the groups, we added comparative statistics to Table 1. We show that there is a difference between the groups with regard to level of education, but this difference is small according to the effect size. Unfortunately, we have no data on socio-demographic background.

3. The ethical framework for the patient group states that patients gave written consent for their anonymized data to be used. There is a statement that no ethical approval is required as data were collected from standard care. That may be the case for the patients, but can the authors please clarify what was the ethical framework for the controls? 

 We clarified the ethical framework of controls by adding a statement under the subheading “Procedure” in the method section on the ethical approval and written informed consent provided by each control subject. 

4. I note that patients with other neurological and psychiatric conditions were included here. These should be excluded from this type of analysis, as other neurological and psychiatric conditions may confound the main question of interest. 

 We chose not to exclude these comorbidities, but to make a critical comment on this in the discussion section under the subheading “Strengths, limitations and recommendations for future research”. We also explain here the reasoning behind this choice. 

5. I also found the results section written in a confusing way. It was not made clear why it mattered that each of the models were a good fit for the data, or what good reliability means for this dataset. 

 We have adjusted the method as requested in comment 1. With that, we hope that the results will also be clearer and more easy to interpret. We attempted to minimize interpretation in the results section. In the discussion section, we elaborate on the meaning of the results. This part we tried to clarify based on your comments. 

6. Further to this, some of the more interesting findings were buried towards the end of the results section. For example, in the PD group, even in patients with no ophthalmic diagnoses, scores were higher on the SVCq. This could be brought out more. 

 We agree that results using the subscale scores are relevant. However, the main aim of the study is to investigate the factor structure. Therefore, we have given results of this analysis first; also because the remaining analyses are a follow-up and based on the factor structure found.

Based on your note, we did add some sentences on the relevance of these follow-up results for clinical practice in the discussion section (subheading “Clinical implications”) and conclusion. 

We also added a small extra analysis on the subscales and the effect of medication use (Levodopa Equivalent Daily Dose; LEDD).

7. It is important that the questionnaire scores increased with age. This should be corrected for in the main analyses. 

 We agree that it is important to take into account that people who are older may develop more complaints when interpreting the questionnaire. We found that this is the case for three of the five subscales. However, correlations were weak. We also found small effects of sex and disease duration on some subscales, and a medium effect of disease severity. We chose not to correct any analysis for effects on other variables.

Furthermore, we have no reason to believe that the factor structure will be different for people of younger age than for people of older age. For example, the factor models found in controls with a lower mean age are the same as the well-fitting models found in older people with PD. Moreover, using a different factor structure for different age groups is not feasible in clinical practice.

8. I also note Plos One policy that data presented in manuscripts should be made openly available. The current statement is that data will be shared upon reasonable request. In order to comply with plos one policy, I would suggest that some form of the data presented here should be made openly available. 

 We will make the data openly available.

Minor comments 

1. Figure 1 is actually a list of questions from the questionnaire and should be changed to a table. 

 We have changed Figure 1 into a table.

2. The abstract is quite hard to follow. Please can the authors clarify what they mean by the different factor models and why this is important to test, in the abstract. 

 We provided a better explanation of the aims of this study in the abstract. Furthermore, we have tried to explain more clearly what the models consist of.

3. Introduction, Page 3, line 41. It is worth mentioning that visual changes are often subtle, and that patients do not always mention them unless specifically asked. And also that the reason it is worth looking for these is that visual changes are predictive of poor outcomes in PD (see e.g. Hamedani et al 2020; Anang 2014). 

 We changed the first paragraph of the introduction section based on these suggestions. 

4. Table 1: please can the authors add comparative statistics and p values as additional columns. 

 Comparative statistics were added to Table 1.

5. It would be good to have the full questionnaire as a table or supplemental element so that the reader can easily find this. Also please can you add information on how long it takes to administer. 

 We added the full Dutch and English questionnaire in the supporting information. In addition, we added a sentence in the method section (subheading “Procedure”) about the time it took to administer the questionnaire.

6. Page 5, line 98 – were data anonymized or pseudo-anonymised? 

 It was pseudo-anonymized, which we corrected in the manuscript. 

7. Please can acronyms e.g. RMSEA, CFA, be explained at first use? 

 We gave an explanation of each of the goodness-of-fit statistics at first use (method section, subheading “Data-analysis”). 

8. Page 8, line 155 “statistics show” – please clarify which statistics. 

 These were the “goodness-of-fit statistics”, which we corrected in the manuscript. 

9. Table 2. Which column is the normed chi square? 

 We added an explanation of the normed Chi-square value in the note of the table (“χ²/df = normed Chi-square”).

10. Tables 4,5, and 6, were these all the 5 factor model? 

 We have placed a sentence directly under the heading "Factor scores and relationships with other variables" indicating that, because the five-factor model had the best fit, all other results are based on that model. 

We also clarified this in the method section (subheading “Factor scores and relationships with other variables”). We stated that subscale scores were calculated for the “five” factors retained from the CFA.

11. Page 15, line 247, the findings of Borm et al, were these in PD or in controls? 

 We clarified that these results applied to “people with PD”.

12. Could mention in the Discussion that these questionnaires help identify patients with PD who have poor visual function and that this could be used to stratify patients and identify those at higher risk for poor outcomes such as dementia. 

 We added a statement on this in the discussion section under the subheading “Clinical implications” and the conclusion.

---

## [Decision Letter · Decision Letter 1]

22 Jul 2022

The Screening Visual Complaints questionnaire (SVCq) in people with Parkinson’s disease - Confirmatory factor analysis and advice for its use in clinical practice

PONE-D-22-10017R1

Dear Dr. van der Lijn,

We’re pleased to inform you that your manuscript has been judged scientifically suitable for publication and will be formally accepted for publication once it meets all outstanding technical requirements. Please not that there are some grammatical errors that have crept in that will need to be addressed prior to formal publication.

Kind regards,

Diego Kaski, PhD MBBS

Academic Editor

PLOS ONE

Additional Editor Comments (optional):

Reviewers' comments:

Reviewer's Responses to Questions

**Comments to the Author**

1. If the authors have adequately addressed your comments raised in a previous round of review and you feel that this manuscript is now acceptable for publication, you may indicate that here to bypass the “Comments to the Author” section, enter your conflict of interest statement in the “Confidential to Editor” section, and submit your "Accept" recommendation.

Reviewer #1: All comments have been addressed

2. Is the manuscript technically sound, and do the data support the conclusions?

Reviewer #1: Yes

3. Has the statistical analysis been performed appropriately and rigorously? 

Reviewer #1: Yes

4. Have the authors made all data underlying the findings in their manuscript fully available?

Reviewer #1: Yes

5. Is the manuscript presented in an intelligible fashion and written in standard English?

Reviewer #1: Yes

6. Review Comments to the Author

Reviewer #1: Thank you for the changes you have made to the manuscript. I appreciate the increased clarity about the analyses, and the modifications to the abstract, methods, and results especially. I also appreciate that you made the data available.

There are a couple of minor grammatical errors that have crept in: eg abstract line 31, an extra “were”; and Methods, Page 4, line 78 “PD over de age groups”.

I am happy for the editor to resolve these with the authors.

7. PLOS authors have the option to publish the peer review history of their article (what does this mean?). If published, this will include your full peer review and any attached files.

Reviewer #1: No

---

## [Editor Report · Acceptance letter]

2 Sep 2022

PONE-D-22-10017R1 

The Screening Visual Complaints questionnaire (SVCq) in people with Parkinson’s disease - Confirmatory factor analysis and advice for its use in clinical practice 

Dear Dr. van der Lijn:

I'm pleased to inform you that your manuscript has been deemed suitable for publication in PLOS ONE. Congratulations! Your manuscript is now with our production department. 

Kind regards, 

on behalf of

Dr. Diego Kaski 

Academic Editor

PLOS ONE